# Systematic Machine Translation of Social Network Data Privacy Policies †

Irfan Khan Tanoli [1,*,‡,§] , Imran Amin [1,§] , Faraz Junejo [1,§] and Nukman Yusoff [2,*,§]

1 Departement of Computer Science, SZABIST, Karachi 75600, Sindh, Pakistan
2 Department of Engineering Design and Manufacture, Faculty of Engineering, University of Malaya, Kuala Lumpur 50603, Malaysia
* Correspondence: dr.irfankhan@szabist.pk (I.K.T.); nukman@um.edu.my (N.Y.)
† This paper is extended version of the paper "*Towards automatic translation of social network policies into controlled natural language*" published in *2018*, *12th International Conference on Research Challenges in Information Science (RCIS)*.
‡ Current address: SZABIST, Karachi 75600, Sindh, Pakistan.
§ These authors contributed equally to this work.

**Abstract:** With the growing popularity of online social networks, one common desire of people is to use of social networking services for establishing social relations with others. The boom of social networking has transformed common users into content (data) contributors. People highly rely on social sites to share their ideas and interests and express opinions. Social network sites store all such activities in a data form and exploit the data for various purposes, e.g., marketing, advertisements, product delivery, product research, and even sentiment analysis, etc. Privacy policies primarily defined in Natural Language (NL) specify storage, usage, and sharing of the user's data and describe authorization, obligation, or denial of specific actions under specific contextual conditions. Although these policies expressed in Natural Language (NL) allow users to read and understand the allowed (or obliged or denied) operations on their data, the described policies cannot undergo automatic control of the actual use of the data by the entities that operate on them. This paper proposes an approach to systematically translate privacy statements related to data from NL into a controlled natural one, i.e., CNL4DSA to improve the machine processing. The methodology discussed in this work is based on a combination of standard Natural Language Processing (NLP) techniques, logic programming, and ontologies. The proposed technique is demonstrated with a prototype implementation and tested with policy examples. The system is tested with a number of data privacy policies from five different social network service providers. Predominantly, this work primarily takes into account two key aspects: (i) The translation of social networks' data privacy policy and (ii) the effectiveness and efficiency of the developed system. It is concluded that the proposed system can successfully and efficiently translate any common data policy based on an empirical analysis performed of the obtained results.

**Keywords:** natural language; controlled natural language; natural language processing; privacy policies; social networks; machine learning

## 1. Introduction

The advent of Online Social Networks (OSNs) allows users to establish and maintain interpersonal relations among people without any boundaries [1]. OSN interactions usually require exchanging users' data for numerous purposes, including the provisioning of services. However, by offering such services for virtual social interaction and data sharing, OSNs also raised user privacy issues by having access to personal data and exposing it, such as blogs, videos, images, or user profile information, e.g., name, date of birth, phone number, email, etc. This shared data leaves traces that may disclose users' activities and their opinions, norms, consent, and beliefs. OSNs usually regulate the collection, usage,

and sharing of users' data (e.g., Facebook [2], Twitter [3], Google [4]), etc. in terms of privacy policies. Usually, the policies [5] published in English describe the terms and conditions under which the provider will manage the data in terms of, e.g., authorization, obligation, or denial. Although the use of English as a natural language enables end-users to read and understand the operations allowed (or obliged, or denied) on their data, a key fact exists that a plain Natural Language (NL) cannot be used as the input language for a policy-based software infrastructure dedicated to automatic policy management and machine readability [6]. Both automated policy analysis (the process to assure the lack of conflicting data policies, see, e.g., [7]) and policy enforcement (the actual application of the data policies, whenever a data access request takes place) require machine-readable language as inputs, such as the standard XACML [8].

An approach proposed for translating Natural Language (NL) data privacy policies [6] as they appear on an OSN website into a Controlled Natural Language (CNL) is the so-called *Controlled Natural Language for Data Sharing Agreements (CNL4DSA)* [9–11]. The system refers to 'Natural Language Policy Translator (Natural Language Policy Translator (NLPT) 1.0)', which outlined the policy translation on a small set of Facebook data privacy policies [6].

This paper presents the extended prototype version of '*Natural Language Policy Translator (NLPT) 1.0*', [6] the so-called '*Natural Language Policy Translator (NLPT) 2.0*', with improvements from the previous version. The system (*NLPT 2.0*) is equipped with a user-friendly Graphical User Interface (GUI), and it is composed of different components, i.e., *Policy Parser (PP)*, *Policy Processor (PR)*, *Ontology Builder (OB)*, *Fragment Extractor (FE)*, *Context Extractor (CE)*, and *Controlled Natural Language (CNLT)* (details of each component is discussed in Section 5). The system (*NLPT 2.0*) simply allows non-expert users to write or input policies in Natural Language (NL) sentences, and the system automatically translates it into CNL4DSA. To validate the system's performance, it is tested with five popular social network platforms' data privacy policies, i.e., Twitter [3], Facebook [2], Google [4], Instagram [12] and LinkedIn [13]. Moreover, the system is also designed to assist researchers in evaluating the specification of social networks' data privacy policies but can also be utilized for the other application domains (e.g., e-health, e-commerce, etc.).

The rest of the paper is outlined as follows: Section 2 gives an overview of the Controlled Natural Languages (CNLs). Section 3 explores the previous literature work. Section 4 describes the proposed methodology. Section 5 depicts the overall architecture of the system. Section 6 presents the experimental setup. Section 7 explains the results and analysis. Finally, Section 8 concludes the work with possible future directions.

## 2. Controlled Natural Language

Generally, formal languages have been proposed and used as knowledge representation languages as they are designed with proper well-defined syntax and unambiguous semantics and support automated reasoning [14]. However, their syntaxes and semantics are quite complex for domain experts to understand and recognize and cause a cognitive distance to the application domain that is not inherent in the Natural Language (NL). One approach to bridge the gap between natural and formal ones is the utilization of Controlled Natural Language (CNL) and full Natural Languages (NLs), developed with well-defined grammar and vocabulary to make statements more understandable and unambigious [15,16]. CNLs have certain writing rules that are, in general, easier for humans to understand and easier for machines to process [14]. CNLs are generally defined in the literature with attributes such as *processable*, *human-readable*, *structured* and *simplified* [17]. Controlled Natural Language (CNL) approaches has evolved in different environments based on the requirements, i.e., different specifications in the context of industry, academia, and government, as well as in different disciplines, i.e., artificial intelligence, computer science, linguistics, biology, literature, etc., since 1930 till today [17].

CNLs have been classified into two broad categories: human-oriented and machine-oriented [18]. Both are built for different purposes and have various applications. Human-oriented CNLs are designed to help humans read and understand technical documents,

e.g., ASD Simplified Technical English [19], and to make human-to-human interactions simpler in certain situations, for instance, air traffic control [20]. Machine-oriented CNLs are designed for the Semantic Web to improve the translation of technical documents and enhance knowledge representation and processing [14]. These languages facilitate the technical document's translatability, e.g., [21], and the acquisition, representation, and processing of knowledge (e.g., for knowledge systems [22] and, in particular, for the Semantic Web [16].

### 2.1. CNL4DSA

CNL4DSA, *Controlled Natural Language for Data Sharing Agreements*, is chosen as a target language. This language was introduced in [9,23], within the EU projects Consequence (Consequence: http://www.consequence-project.eu/) and Coco Cloud (Coco Cloud: http://www.coco-cloud.eu/) and successfully applied to the pilots of the two projects. It is equipped with analytic tools, i.e., a policy authoring tool, a policy analyzer, and conflict solver, and a policy mapper of enforceable language. CNL4DSA was originally developed for editing so-called *data sharing agreements* (formal contracts regulating data sharing), allowing a simple and readable, yet formal, specification of different classes of privacy policies, as listed below:

- **Authorizations**, referring to permission for subjects to perform actions on object under a specific context.
- **Prohibitions**, expressing the fact that a subject cannot perform actions on an object under a specific context.
- **Obligations**, referring to subjects obliged to perform actions on objects under a specific context.

CNL4DSA relies on the notion of *fragments*, tuples of the form $f = \langle s, a, o \rangle$, where $s$ is the subject, $a$ is the action, $o$ is the object. A fragment simply says that 'subject $s$ performs action $a$ on object $o$'. By adding $can/must/cannot$ constructs to the basic fragment, a fragment becomes either an authorization, obligation, or prohibition. In the scenario of social networks, subjects are usually physical or legal entities (e.g., users and service providers), actions are, e.g., collect, login, etc., while objects consist of any data published on the social network or stored on its servers, e.g., the personal details of a Facebook account, content created by the social network users, or the data policies themselves.

Fragments are assessed in a certain *context*. A context is assessed as a Boolean value (true/false) in CNL4DSA. It makes claims about the characteristics of subjects and objects in words such as user's roles, data categories, date, and location. Simple context examples are 'subject hasRole Facebook_admin', or 'object hasCategory user_post'. It predicates the constructs that connect subjects and objects to their values, such as hasRole and hasCategory in the examples above. Contexts must be mixed to define complicated policies. Therefore, the Boolean connectors *and*, *or*, and *not* are used to indicate a composite context $C$, which is defined inductively as follows :

$$C := c \mid C \text{ and } C \mid C \text{ or } C \mid \text{not } c$$

The syntax of a *composite fragment* denoted as $F_A$ is as follows:

$$F := nil \mid can, must, cannot \ f \mid F; F \mid if \ C \ then \ F \mid after \ f \ then \ F$$

- *nil* can do nothing.
- *can, must, cannot f* is the atomic fragment that expresses that $f$ is allowed/require/not permitted, where $f = \langle s, a, o \rangle$. Its informal meaning is *the subject s can perform action a on the object o*.
- *F; F* is a list of composite fragments (i.e., a list of authorizations, obligations, or prohibitions).
- *if C then F* expresses the logical implication between a context $C$ and a composite fragment: if $C$ holds, then $F$ is allowed/required/not allowed.
- *after f then F* is a temporal sequence of fragments. Informally, after $f$ has happened, then the composite fragment $F$ is allowed/required/not allowed.

Additionally, the syntax used by CNL4DSA to represent composite obligation and prohibition fragments are unique. The obligation fragment, such as the authorizations, states that *the subject s must perform action a on the object o*, whereas *the subject s cannot execute action a on the object o* is stated for the prohibition.

### 2.2. Scenario Examples

Consider the example of an emergency situation in which many cars are engaged in a collision, including a tanker [23]. Firefighters, Red Cross paramedics, and toxicologists all rush to help the wounded. Firefighters and Red Cross volunteers are referred to as 'rescuers' or 'Rescuers' in a general sense (complete details available in [23]).

Consider the following sample cases:

1. P1: Firefighters can access the victim's personal and medical information.
2. O1: Once the alert states of the accidents have been determined by the Red Cross members, if it is larger than five, they must then inform the local community of the alert level.
3. PT1: Non-firemen cannot access tanker delivery notes that are currently in progress.

*The expression of P1 in CNL4DSA are as follows:*

**IF c THEN CAN f**

where:

- c = hasRole(user1, fireman) and hasDataCategory(data, personal) and hasDataCategory(data, medical) and isReferredTo(data, user2) and isInvolvedIn(user2, accident) is a composite context.
- f = can access(user1, data) is a composite authorization fragment.

*The expression of O1 in CNL4DSA are as follows:*

**IF c THEN MUST f**

where:

- c = hasRole(user1, RedCross) and hasDataCategory(data, alertState) then after that access(user1, data) then if isGreaterThan(alertState,five).
- f = must communicate(user1,data) is a composite obligation fragment.

*The expression of PT1 in CNL4DSA are as follows:*

**IF c THEN CANNOT f**

where:

- c = not hasRole(user1,fireman) and hasDataCategory(data, deliveryNote) and isReferredTo(data,truck) then cannot access(user1, data) where not hasRole(user1, fireman) and hasDataCategory(data,deliveryNote) and isReferredTo(data,truck) is a composite context.
- f = cannot access(user1, data) is a composite prohibition fragment.

The impetus for selecting CNL4DSA as the aim translation's language is due to the following facts: first, as proposed in [9], composite fragments have formal semantics that are described by *modal transition systems*. Because of this, the language may be formally analyzed, even using already available tools such as *Maude* [24]; for instance, the authors of [7,25] show the automated analysis of data sharing and privacy policies completely; second, the CNL4DSA is equipped with an editor with a dedicated authoring tool, having preloaded domain-specific vocabularies in the form of ontologies, understandable for machine translation, and can be automatically mapped into a low-level language, namely XACML [8], which enables seamless policy enforcement.

## 3. Related Work

Controlled Natural Languages (CNLs) are generally discussed in the literature with characteristics such as *easy to process*, *human-readable*, *structured* and *simplified* [17]. Controlled Natural Languages (CNLs) are more contrived subsets of Natural Languages (NLs), whose representation—including grammar, syntax, semantics, and vocabulary—have been developed in a simple but more efficient way to reduce and mitigate the ambiguity and complexity of natural language and make it feasible for machine processing [14]. A variety of CNLs have been proposed for different purposes by researchers of diverse expertise and background [17], e.g., Tateish et al. [26] propose an approach to automatically generate a smart contract from a Natural Language (NL) contract document that is defined using a document template and a Controlled Natural Language (CNL). The system is based on the mapping of the document template and the Controlled Natural Language (CNL) to a formal model that can describe the terms and conditions in a contract, including temporal constraints and procedures. The formal model is translated into an executable smart contract. A framework for tax fraud detection was developed by Calafato et al. [27], where the fraud expert is empowered to design tax fraud patterns through a Controlled Natural Language (CNL) independently. Colombo et al. [28] suggested a Controlled Natural Language (CNL) that maintains machine readability while allowing non-technical users to make queries that are simple to comprehend.

The automatic and unambiguous translation from a Controlled Natural Language (CNL) to first-order logic is demonstrated by Fuchs et al. [22]. Originally intended to be a specification language, the language has evolved to focus on knowledge representation and applications for working with the Semantic Web [17]. A tool known as 'RuleCNL (a CNL for creating business rules) incorporates formal syntax and semantics to enable business professionals to formalize their business rules in a business-friendly manner that machines can understand [29]. The key feature of 'RuleCNL' is the business rule definition alignment with the business vocabulary that assures consistency and traceability in this domain.

Brodie et al. [30] developed a policy workbench called 'SPARCLE' for parsing privacy policy rules in Natural Language (NL). 'SPARCLE' enables organizational users to enter policies in Natural Language (NL), parse the policies to identify policy elements, and then generate a machine-readable Extensible Markup Language (XML) version of the policy. The work is empirically evaluated from the usability perspective by targeting organizational privacy policies. This results in the successful implementation of the parsing capabilities to provide a usable and effective method for an organization to map the natural language version of privacy policies to their implementation and subsequent verification through compliance auditing of the enforcement records.

Fisler et al. [31] provided an authoring language based on Datalog-like formats as input for the policy editor. The work emphasizes the social and environmental aspects that can influence the interpretation and specification of trust and privacy policies. Kiyavitskaya et al. [32] proposed a methodology that delivered the transformation of natural language into semi-structured specifications. The approach suggests a mechanism to support designers during requirements elicitation, modeling, and analysis.

Fantechi et al. [33] provided a tool that formalizes the behavioral needs of reactive systems into a process algebra and converts natural language phrases into ACTL (Action-based Temporal Logic). A logic-based framework for policy analysis was developed by Craven et al. [34] that enables the expression of responsibilities and authorizations offers practical diagnostic data and allows for dynamic system modeling. Fockle et al. [35] created a model-driven development-based methodology for requirements engineering. For documentation, elicitation, and requirements negotiation, the system depends on requirements models and a controlled natural language. Through a bidirectional, multi-step model transformation between two documentation forms, the methodology combines the advantages of model-based and natural language documentation.

A graphical visual interface with an adequate level of abstraction was developed by Mousas et al. [36] to allow users to specify fundamental ideas for privacy protection, such

as values for roles, activities, data kinds, rules, and contextual information. Ruiz et al. [37] developed a software infrastructure to handle data sharing agreements (DSA), which govern data access, usage, and sharing. The framework permits DSA editing, analysis, and enforcement. The input language used by the authoring tool to edit the DSA is 'CNL4DSA' [9]. The CNL4DSA provides easy yet explicit specifications for many privacy policy kinds, including authorizations, obligations, and prohibitions. The language can be automatically mapped into a low-level language, XACML [8], thus enabling seamless policy enforcement.

The approach proposed by Tanoli et al. [6] is one of the few initiatives to bridge the gap between modifying and processing a Controlled Natural Language (CNL), such as CNL4DSA, and maintaining complete readability of privacy policies by expressing them in a natural language. The approach relies on standard and well-established Natural language Processing (NLP) techniques, e.g., Crossley et al. [38], the Natural Language Toolkit [39], the Stanford CoreNLP [40], Spacy [41] and the adoption of ontologies [42].

## 4. Design Approach

This section highlights an overview of the design approach for semi-automatically translating natural language data privacy policies into CNL4DSA. A prototype console-based system referred to as '*Natural Language Policy Translator (NLPT) 1.0*' is designed to translate the data policy into CNL4DSA [6]. The core of the approach relies upon three phases: (i) Natural Language Processing (NLP); (ii) building ontologies; (iii) translation into CNL4DS using logic programming. The various steps are depicted in Figure 1. (The complete details following step-by-step policy translation, as shown in Figure 1, are discussed in [6]).

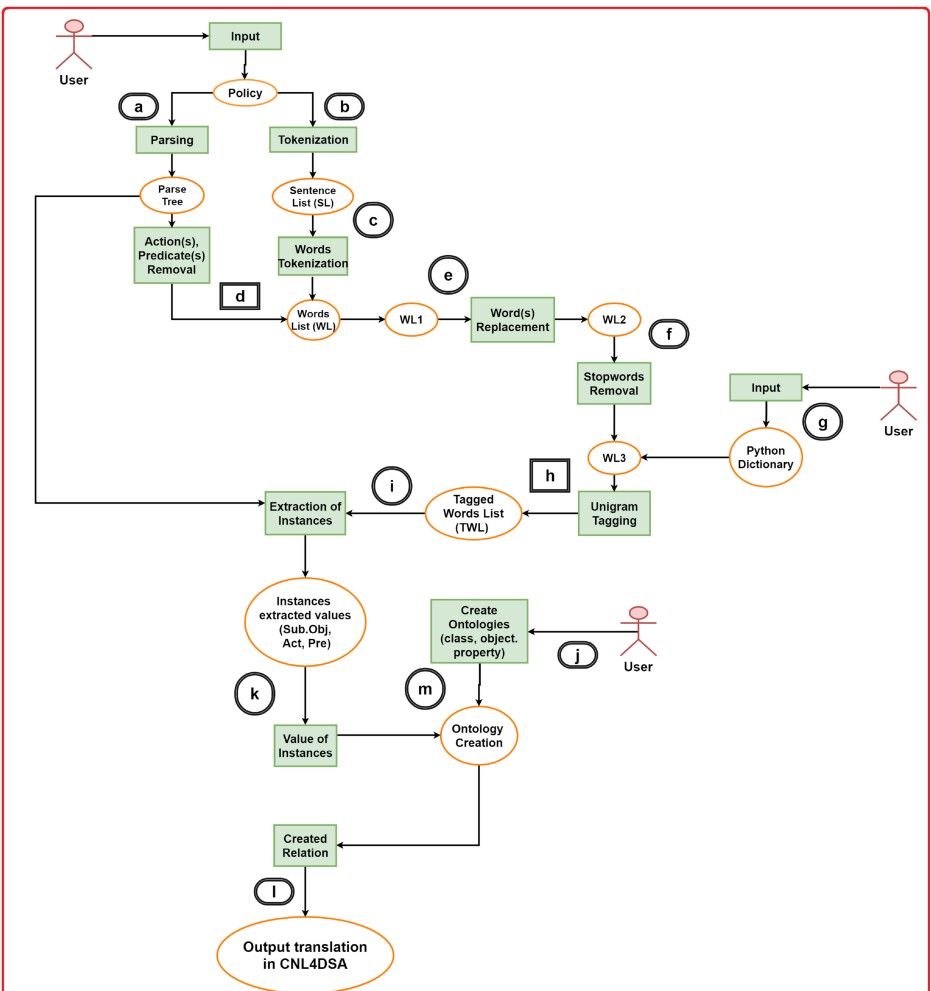

**Figure 1.** Pictorial representation of the system operations.

*4.1. Policy Translation Using Natural Language Policy Translator (NLPT) 1.0*

To explain the process of three phases, consider the following subset of the Facebook data privacy policy as a case example—available at [43].

*P1: "We collect the content and other information you provide when you use our Services [...]"*

Policy P1 is an authorization policy, allowing the social networking service provider to collect information provided by the user when the user interacts with the platform services.

P1 as an authorization policy in CNL4DSA:

The CNL4DSA representation of the policy P1:

**IF** c1 **THEN CAN** f1

where:

- c1 = subject1 hasRole '*social_networking_service_provider*' AND (object1 hasCategory '*content*' OR object1 hasCategory '*other_information*') AND (subject2 provides object1 OR subject2 provides object1) AND subject2 hasRole '*user*' AND subject2 uses '*social_networking_service*' is a composite context.
- f1 = subject1 collect object1 is an atomic fragment.

In the above translation, for the sake of a more direct understanding, the term '*we*' of the original policy is substituted with '*social_networking_service_provider*', the term '*you*' with '*user*', and the term '*service*' with '*social_networking_service*'.

4.1.1. Natural Language Processing

The first phase of the translation process is the application of Natural Language Processing (NLP) techniques to policy P1. The goal of this phase is to automatically derive, from the sentences in Natural Language (NL), the standard elements of a data policy (i.e., subject, object, and action), as well as the typical constructs of CNL4DSA (i.e., fragments and contexts).

Following standard NLP approaches, a data policy is first parsed and then represented in a tree form, using a syntactic dependency parser (step (a) in Figure 1). A dependency parser shows syntactical dependencies among individual words in a sentence, as well as among the main sub-sentence and its subordinates. For our goals, the dependency parser is used to discriminate among actions and predicates for those cases in which the predicate in the sentence is under a verbal form. Such concepts are illustrated with the following example.

Considering P1, it is needed to automatically verify whether 'collect' is the action element of the policy (as it is, see fragment f1 defined in the CNL4DSA representation of P1) or part of a context. To meet the goal, the dependency parser is used. This enables picking up the verb(s) of the main sentence and labeling them as action(s), while the verb(s) in the subordinate's sentences will be tagged as predicates. In this case, the system recognizes the terms 'provide' and 'use' as predicates. Finally, it is worth noting that other predicates, e.g., 'hasRole' and 'hasCategory', will be defined in a subsequent phase during the definition of ontologies (Section 4.1.2).

Upon considering different parsers, i.e., the Stanford parser [44], the one provided by the NLTK toolkit [39], and the one provided by the SpaCy Python package [41], the latter is found to be fast and accurate enough to parse the sentences. Figure 2 shows the resulting tree after applying the SpaCy dependency parser to P1.

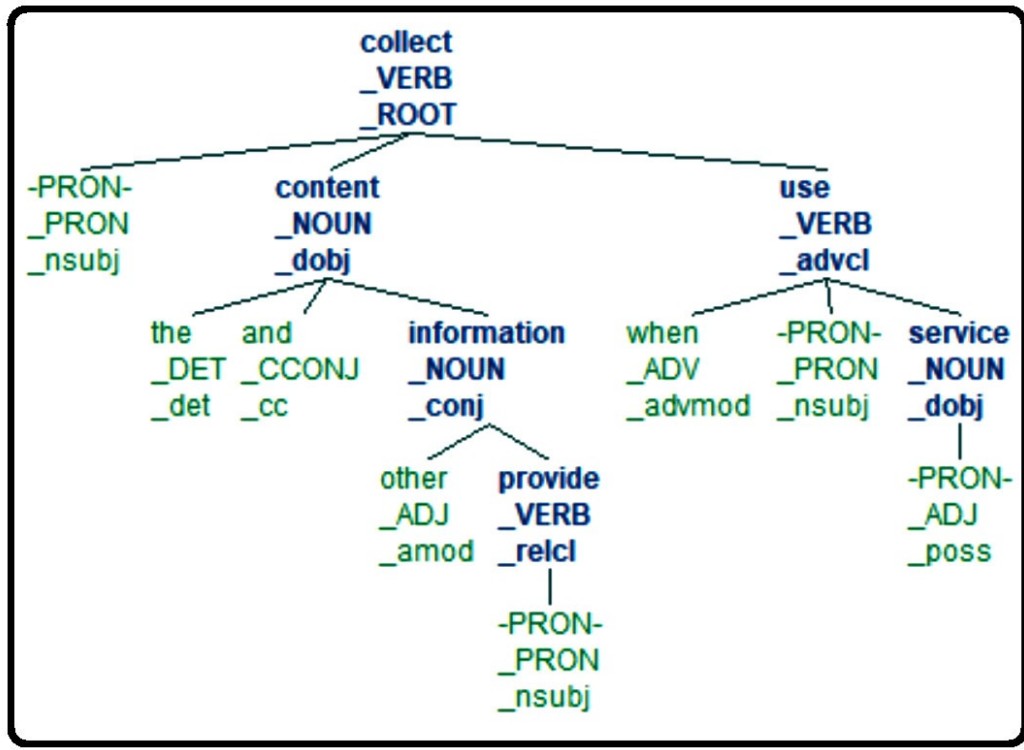

**Figure 2.** P1 upon dependency parser processing.

Referring to Figure 1, the data policy is also split into single sentences, using a sentence tokenizer—step (b). Each sentence is further divided into words, using a word tokenizer—step (c). Plurals are substituted with their singular forms, and different forms of verbs are led back to their first form using the NLTK lemmatizer [39]. Upon step (c), a word list WL is obtained.

In the specific case of P1, the NLTK word tokenizer and the SpaCy dependency parser treat the two terms 'other and information' as two separate words (see, e.g., Figure 2). For the sake of the next processing phase, the two terms have been then manually replaced with a single one '*other_information*' by relying on the Python string replace method [45].

```
Sentences are tokenized:
['we collect the content and other_information
you provide when you use our services]

Words are tokenized:
['we', 'collect', 'the', 'content', 'and',
'other_information', 'you', 'provide','when',
'you', 'use', 'our', 'service']
```

From WL, the terms that constitute the actions and the predicates are removed, as identified by the dependency parsing operations, as well as duplicate items if there are any in a list. The actions and predicates removal corresponds to step (d) in Figure 1. In the second list of words, WL1 is obtained.

```
Action and Predicates removal:
['we', 'the', 'content', 'and','other_information',
'you','when', 'our', 'service']
```

In next phase of word replacement, in which in WL1 is replaced with a few ambiguous words with a more precise meaning. In P1, for example, the term 'service' is replaced with 'social_networking_service', 'we' with 'social_networking_ service_provider', 'you' with 'user'. The replacement happens according to a manually pre-defined list of words: the tool replaces such words accordingly. It is worth noting that, in the prototype implementation

presented in the work [6], automatic co-reference detection is not considered. Although a current limitation of the approach, it is argued that this necessary refinement can be applied in future work, possibly using existing tools for automatic co-reference detection. A modified list of WL2 is obtained.

```
Words replacement with meaningful words:
['social_networking_service_provider','the',
'content', 'and', 'other_information','user',
'when', 'our', 'social_networking_service']
```

Then, a list of stop words is used to remove words, such as articles *an*, *a*, *the*, etc., prepositions, e.g., *in*, *on*, etc., and adverbs, e.g., *before* (step f). As words for removal, the pre-defined NLTK stop words list [39] is applied. The modified list WL3 is obtained.

```
Stop words removal:
['social_networking_service_provider','content',
'other_information','user','social_networking_
service']
```

As already discussed at the beginning of Section 4.1.1, it is required to identify each data policy:

1. The subjects, the objects, and the actions;
2. The contexts and fragments, as required by the CNL4DSA language.

It is worth mentioning that actions and (part of) predicates are already identified during the application of the dependency parser. The Unigram Tagger [46] is exploited to label the tokenized words in WL3 as subjects or objects. The UnigramTagger class implements a simple statistical tagging algorithm: for each token, it sets the most likely label for that type of token. For example, it assigns the 'JJ' (adjective) tag to any occurrence of the word 'frequent' since it is mostly used as an adjective (e.g., *a frequent word*) rather than as a verb (e.g., *I often frequent this place*).

Before actually using the Unigram Tagger to tag the data, it must be trained on the tagged Python dictionary. The creation of the training dictionary happens once, on a set of initial words. For subsequent words, the user will update the dictionary with the possibly encountered new terms, which are not in the original set of terms, until no major update is needed. The tool prompts a message for the user to define a Python dictionary tagged according to a privacy policies terminology—step (g). The user can define as many terms as possible so that the tagging machine can automatically label the words that will appear in subsequent data policies. The use of the Unigram Tagger in this work is specific to tag terms as either subject or object. In addition, the tagger also considers the terms expressing authorizations, prohibitions, and obligations (such as *can*, *must*, and *cannot*).

```
#############################################
Do you want to define a training dictionary?
'Y' or 'N' => Y
Please define a Python dictionary, e.g.,
'user':'subject', 'data':'object'
#############################################
Please proceed with dictionary (subject, object,
as keys) => 'social_networking_service_provider':
'subject','user':'subject','other_user':'subject'
Dictionary Data =>
{'social_networking_service_provider':'subject',
'user':'subject','other_user':'subject'}
#############################################

#############################################
Updated Dictionary =>{'other_information':'object',
'device':'object','data':'object', 'information':
'object','user':'subject','must':'obligation',
'content': 'object','can_not': 'prohibition',
'can':'authorization',
'social_networking_service':'object',
'other_user': 'subject',
'social_networking_service_provider':'subject'}
```

After running the Unigram Tagger on P1—step (h), the following list of tagged words (TWL) is obtained:

```
Words tagged according to policy elements:
[[('social_networking_service_provider','subject'),
('content', 'object'),('user', 'subject'),
('other_information','object'),
('social_networking_service', 'object')]]
```

Regarding the distinction among authorizations, prohibitions, and obligations, the following simple procedure is adopted. The keywords *can, should, may* in the original Natural Language (NL) statement lead to considering the policy an authorization. The keywords *can not, should not, shall not, must not* (and relative contracted forms) lead to considering the policy a prohibition. Finally, the keywords *must, shall, will* characterize obligations. The choice is made following the interpretation given in RFC2119 [47].

Additionally, terms *should* and *may* are replaced with *can* during the replacement phase. Similarly, *shall not should not*, *must not* are replaced with *cannot*, while *shall* and *will* are replaced with *must*. Whenever such keywords appear in the original policy statement, the tool labels them accordingly. However, a current limitation of the approach is that, if such keywords do not appear in the natural language statement, the tool treats the statement as an authorization policy. This is exactly what happens in the case of P1.

Finally, to handle incorrect tagging, when the user defines the Python dictionary, the allowed keys for tags are only: subject, object, authorization, prohibition, and obligation. When more than one term is tagged as subject (resp., as object), the labels are subject1, subject2, . . . , subjectn (resp., object1, . . . , objectn). The same holds for actions and predicates when applying the dependency parser.

Regarding the formation of the CNL4DSA contexts, the next Section 4.1.2 will show how to link objects and subjects to predicates employing ontologies.

4.1.2. Building Ontologies

An ontology is a explicit formal description of a domain of interest [48]. A specific ontology-based vocabulary is defined, inherent to the scenario of privacy policies, which defines terms representing, e.g., categories for objects (such as posts, content, picture, etc.), roles for subjects (such as the user, social networking service providers, Facebook provider, etc.), identifiers for subjects and objects (e.g., John Doe, pic12345) and terms for actions (such as read, send, access, store). Then, the ontology defines the relations between all the terms in the vocabulary. Relations are established using predicates, e.g., hasRole, hasCategory, isTime, hasLocation, etc. Owl ontologies [49] and Owl ready [42], a Python module, are used to load them.

The classes: subject, object, action, category, and role are defined. Below is an example of class declaration—step (j) in Figure 1.

```
class Category(ObjVocabItem):
ontology=onto
```

The predicates are hasRole, hasCategory, provide and use. In Owl ready, these predicates are called object properties. The object properties create relations between the classes. Examples of object properties are:

```
class hasCategory(ObjectProperty):
domain=[Object], range=[Category]
```

or

```
class hasRole(ObjectProperty):
domain=[Subject], range=[Role]
```

As an example, hasCategory is an object property with the domain class Object and range class Category. Moreover, the predicates hasRole and hasCategory are created manually through ontologies, while the predicates use and provide are obtained by the

application of the dependency parser. As a remark, hasCategory and hasRole are object properties of the ontology classes. In Owlready, they are declared with the following syntax: 'class hasCategory (ObjectProperty)', where ObjectProperty syntax refers to the predicate as object property [42]. 'Collect' is defined as a functional property; it has only one value, and it is created by inheriting the FunctionalProperty class [42]:

```
class collect(FunctionalProperty):
domain=[Collect], range = [str]
```

The relations are defined as follows: subject1 hasRole: 'subject1', subject2 hasRole: 'subject2', object1 hasCategory: 'object1','subject2' provide 'object1', 'subject2' provide 'object1', and subject2 use 'object3'.

Figure 3 shows the ontology representation of P1, created using Protégé [50], where Thing is the main ontology class and Term, Action, and ObjVocabitem are the ontology subclasses. Furthermore, Term has Subject and Object as subclasses and Objvocabitem has Category, Subject, Object, Role, etc. To establish relations between subject and object, object properties are used. As an example, to establish a relation between subject and role class, the hasRole object property is used.

To conclude this step, the actual values of subject1, subject2, object1, object2, and object3, as well as the values for action(s) and predicate(s), are required: the subject and object values are extracted from the policy tagged-tokens and create the instances—step (i). Actions and predicates values are extracted from the dependency parsing tree, as follows:

```
Extracted Values for Subject, Action, Object,
Predicate: Subject1 is: ['social_networking_service_provider']
Subject2 is: ['user'], Object1 is: ['content'] Object2 is:
['other_information'],
Object3 is: ['social_networking_service']
Action1 is: ['collect'], Predicate1 is: ['provide']
Predicate2 is: ['use']
```

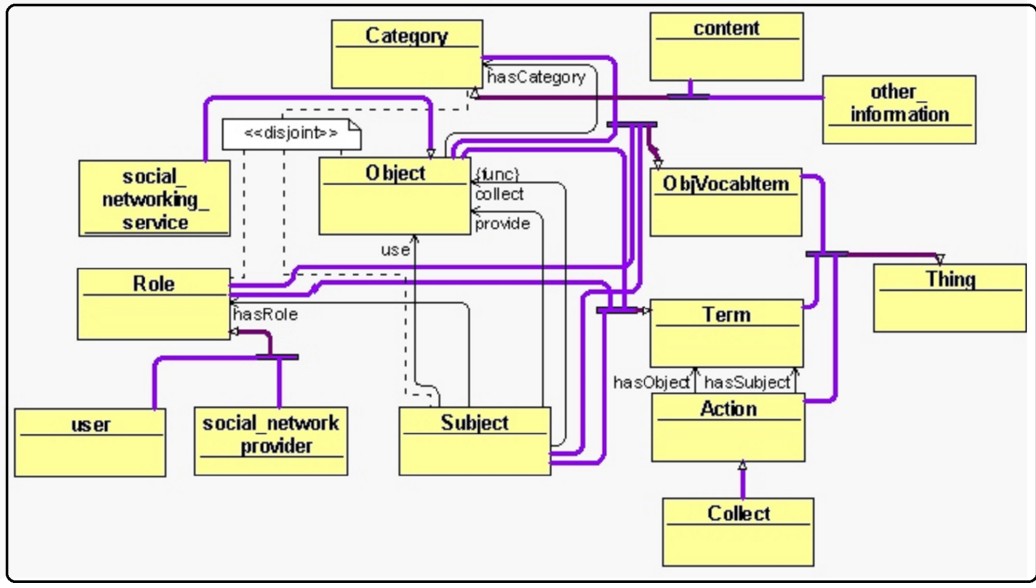

**Figure 3.** Ontology representation for P1.

### 4.1.3. Translation into CNL4DSA

In the final phase, the extracted values for the various instances of subject and object are passed as arguments to the ontology classes previously defined—steps (k), (l), (m). As an example, subject1, with value '*social_networking_service_provider*', is passed to the Role class, and the relation between the subject and the Role class is established through *has_Role*; object1, with value *content*, is passed to the Category class, and the relation between the object and the Category class is established through *has_Category*.

```
print ('IF subject1 hasRole:',Subject1.hasRole)
output: IF subject1 hasRole: [onto.'social_networking_service_provider']
```

Similarly, the action 'collect' is passed as an argument to the Action class. The printed output represents the translation of the natural language data policy into CNL4DSA. The outputs are stored in variables c1 and f1:

$$IF\ c1\ THEN\ CAN\ f1$$

where:

```
c1 = subject1 hasRole: [onto.social_networking_service_provider]
AND (object1 hasCategory: [onto.content] OR object2 hasCategory:
[onto.other_information]) AND subject2 hasRole: [onto.user] AND
(subject2 [onto.user] provide object1:[onto.content] OR
subject2 [onto.user] provide object2 [onto.other_information])
AND subject2 [onto.user] use [onto.social_networking_service]
is a composite context.
f1 = subject1 (action): [collect] object1 is an atomic fragment.
```

*onto* is the Python variable that is used to modify, save, and load ontologies, for example: onto = *get_ontology* [42]. The example of obligation and prohibition policy is presented and discussed in the work [6].

## 5. Architecture and Graphical User Interface

In Section 4, the mechanism of the social network data privacy policy translation into CNL4DSA is demonstrated. Initially, the system was designed to translate only a limited set of Facebook data privacy policies [2]. Here, an improved new prototypical version of *Natural Language Policy Translator (NLPT) 1.0* is referred to as *Natural Language Policy Translator (Natural Language Policy Translator (NLPT) 2.0)*. The major development in the *NLPT 2.0* is that it is equipped with a user-friendly graphical interface to directly input the policy, and the system is capable enough to translate any common social network data privacy policies defined in Natural Language (NL) into CNL4DSA.

*Natural Language Policy Translator (NLPT) 2.0* is composed of the following components:

- *Policy Parser (PP)*
- *Policy Processor (PR)*
- *Ontology Builder (OB)*
- *Fragment Extractor (FE)*
- *Context Extractor (CE)*
- *Controlled Natural Language Translator (CNLT)*

The translation of the policy is split among these components to ease the user into understanding the translation process, which was not possible in the previous version ('Natural Language Policy Translator (NLPT) 1.0').

The high-level architecture of the system '*(Natural Language Policy Translator (NLPT)) 2.0*' is presented in Figure 4.

The detailed working of each component is as follows:

**Policy Writer:** A person or entity who properly enters the policy is known as the "Policy Writer". To prevent potential incorrect/wrong tagging performed by the policy parser, a policy writer is essential. The *Policy Writer* just needs to be an end-user who is aware of the proper way to write/enter the policy in the system. They do not need to be a Controlled Natural Language (CNL) expert or a domain specialist. A *Policy Manual* with all essential instructions on how to write/input the policy is provided to train the *Policy Manual*. The *Policy Manual* is a document that outlines a process for properly combining Subject, Verb, and Object (SVO) in an original policy by giving various policy examples [51].

The introduction of the *Policy Writer* role is diverse since writing styles are used by online service providers to describe the policies, and if a privacy statement is lengthy and complex, it is possible that it may contain several distinct predicates. As a result, it is possible that the dependency parser incorrectly tags some policy terms or fails to

appropriately parse the statement. These problems also directly affect the translation of policy, especially in terms of the extraction of actions and predicates.

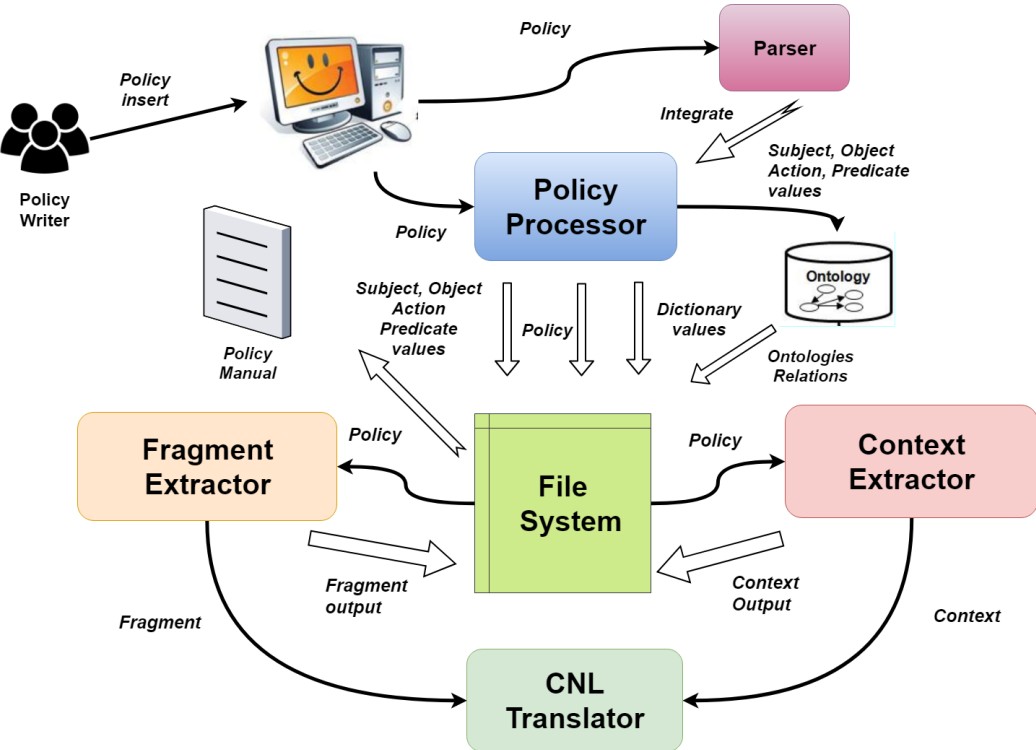

**Figure 4.** System architecture *Natural Language Policy Translator (NLPT) 2.0*.

Consider the following Facebook policy example:

**Policy M1**: 'We also collect contact information if you choose to upload, sync, or import it from a device [...]' [43]. .

When the *Policy Writer* inputs M1 in *Natural Language Policy Translator (NLPT) 2.0*, and upon processing M1 using the policy parser, the following output is generated, as shown in Figure 5.

The Policy Parser (PP) labels 'sync' as a noun, but this should be tagged as a verb. However, the policy writer can rephrase the policy following policy manual guidelines to specify it properly, i.e., the parser tags the words properly with the right combination of Subject, Verb, and Object (SVO).

An adequate rephrasing is:

**Rephrase M1:** 'We also collect data if you choose_to _upload data or you sync data, or you import data from a device.

When using the system once on M1, it is apparent that with the right rephrasing, the parser correctly tags 'sync' as a verb, as demonstrated in Figure 6. It is important to note that for better and more precise tagging, words such as 'information', 'content', 'contact information, 'content and information', etc., are replaced with the general term 'data'. An infinitive form verb, e.g., 'decide to signup', (_) is introduced, i.e., 'choose_to _upload', which makes the *Natural Language Policy Translator (NLPT) 2.0* consider infinitive verb(s) as a single word.

**Policy Parser (PP)**: The policy parser simply parses the policy using the Spacy dependency parser [41] to obtain action and predicate(s), as already properly explained in Section 4.

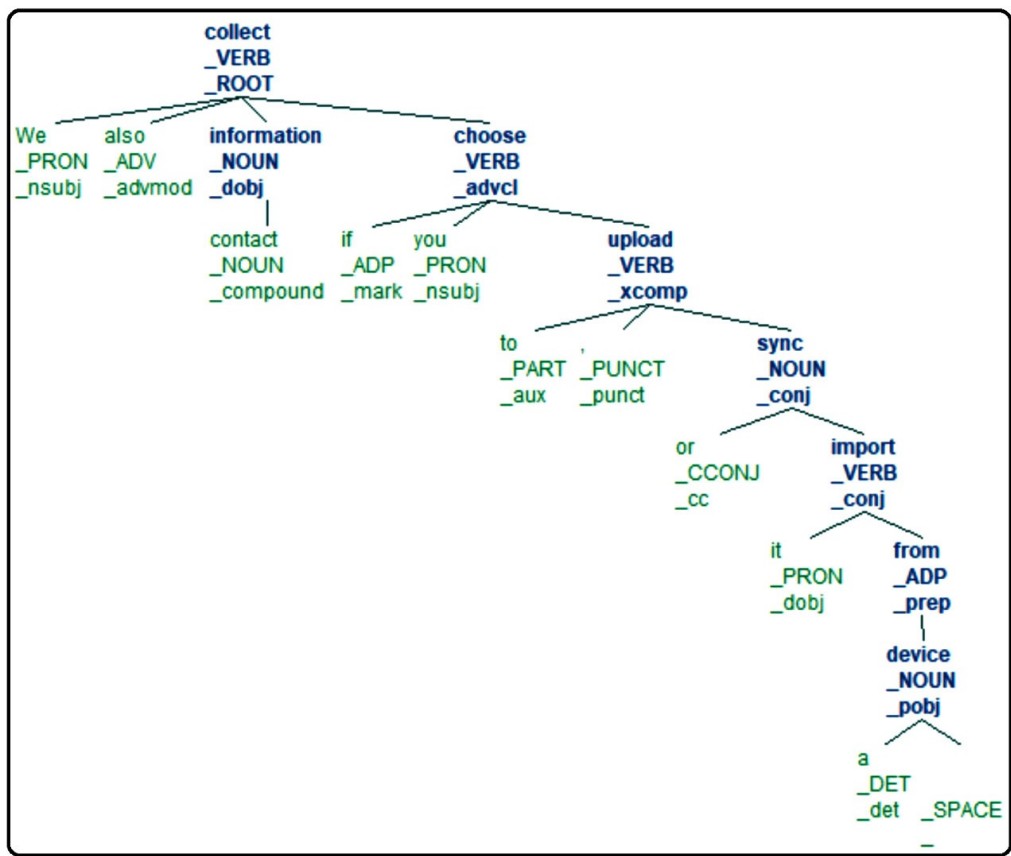

**Figure 5.** Policy (M1) upon dependency parser processing.

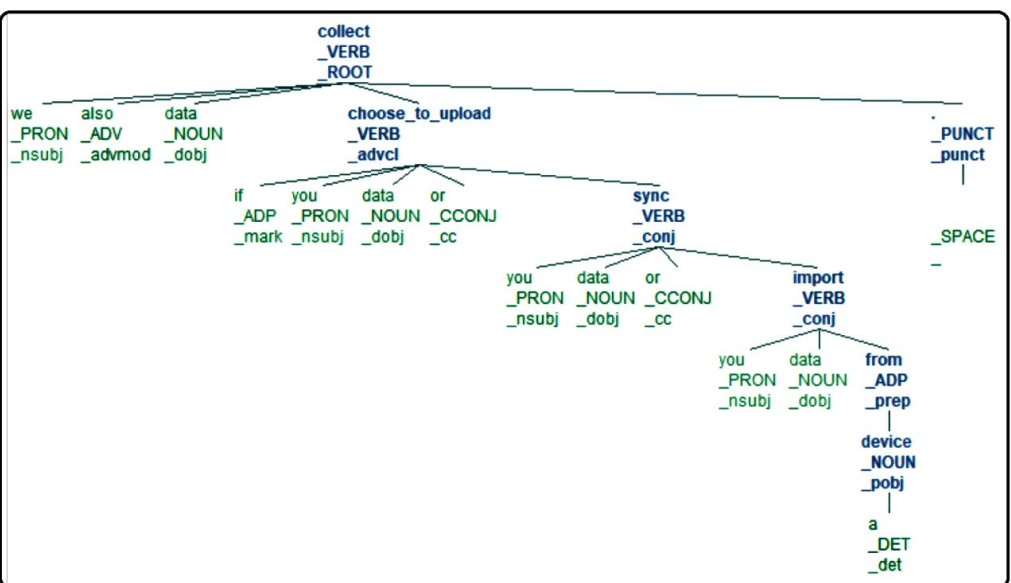

**Figure 6.** Rephrase-policy (M1) upon dependency parser processing—3.

**Policy Processor (PR)**: Policy processor (PR) applies sentence and word tokenization, removes stop-words, and creates a dictionary for extra stop-words. For uni-gram tagging, it allows creating the dictionary containing subject(s) and object(s) in the policy. *PR* is entirely developed with the combination of NLTK [39], and Spacy [41]. The results of this processing are then saved in a file using the Python OS.System function.

**Ontology Builder (OB)**: The identified subjects, objects, actions, and predicates are provided as input to the *Ontology Builder*, which is already furnished with some pre-defined

predicates, such as 'object/subject isRelatedto', 'object/subject isPartof', and 'object/subject hasCategory', for objects, and 'subject hasRole, hasID', and "object hasOwner'. The subject, object, action, category, predicate, id, owner, and role classes are all specified. Owlready, a python package to load ontologies is utilized to implement the *Ontology Builder*. Each time a new predicate is needed to process the policy, the ontology vocabulary is manually updated. The *Context Extractor (CE)* will use the result after storing it in a file.

**Fragment Extractor (FE)**: 'Subject Action Object' is a policy fragment that is extracted by *Fragment Extractor (FE)*. For fragment recognition, this component is implemented in a policy using logic programming; specifically, it translates the subject, action, and object to the values determined by *Policy Processor (PR)*. For instance, the *FE* extracts the fragment *'we collect data': 'subject action object'*. The result is once more recorded in a separate file, saved to the file system, and utilized by the *Controlled Natural Language Translator (CNLT)* afterward.

**Context Extractor (CE)**: The CNL4DSA context is extracted from the policy by *Context Extractor (CFE)*. The output of the *Ontology Builder* is also recalled to obtain any contexts with the predicates 'hasRole' and 'hasCategory'. *Fragment Extractor (FE)* and *Context Extractor (CFE)* are created using logic programming. Concerning policy M1, the anticipated output is:

*subject hasRole: 'we' AND subject hasRole: 'you' AND object hasCategory: 'data' AND subject (you) predicate (choose_to_upload) object (data) AND subject (you) predicate (sync) object (data) AND subject (you) predicate (import) object (data) is a composite context.*

The acquired output is saved once more in a different file and utilized later by the *Controlled Natural Language Translator (CNLT)*.

**Controlled Natural Language Translator (CNLT)**: The outputs of *Fragment Extractor (FE)* and *Context Extractor (CFE)* are simply retrieved from the file system by *Controlled Natural Language Translator (CNLT)*, which then displays the policy translation in CNL4DSA. *Controlled Natural Language Translator (CNLT)* can also categorize the obtained output as authorization, obligation, and prohibition fragments.

**Complete Translation into Controlled Natural Language (CTCNL)** is developed with a mechanism that orchestrates the components to process the translation all at once. It does this by first calling the *Policy Parser (PP)*, then the *Policy Processor (PR)*, *Ontology Builder (OB)*, *Fragment Extractor (FE)*, *Context Extractor (CFE)*, and *Controlled Natural Language Translator (CNLT)*. The full process flow of the translation is depicted in Figure 7 as a single iteration.

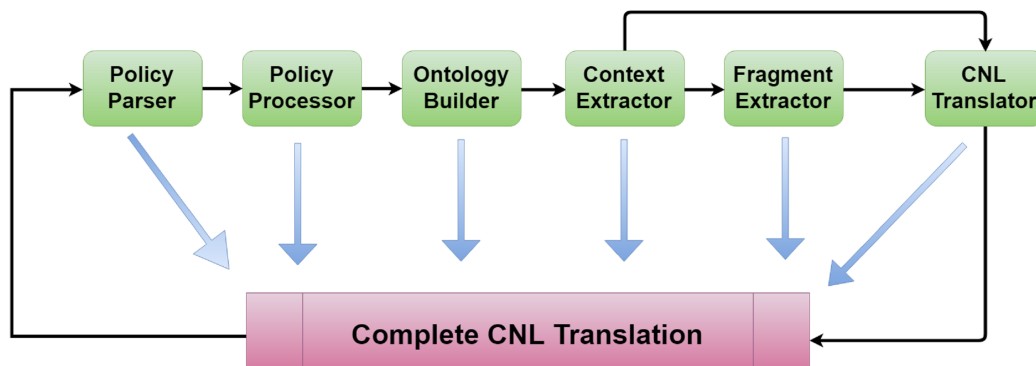

**Figure 7.** Complete Controlled Natural Language (CNL) translation process.

Figure 8 display the Graphical User Interface of *Natural Language Policy Translator (NLPT) 2.0*. The *Graphical User Interface (GUI)* is developed with Python GUI Generator (PAGE) [52]. The are shown on the window's right and bottom sides (in the form of buttons).

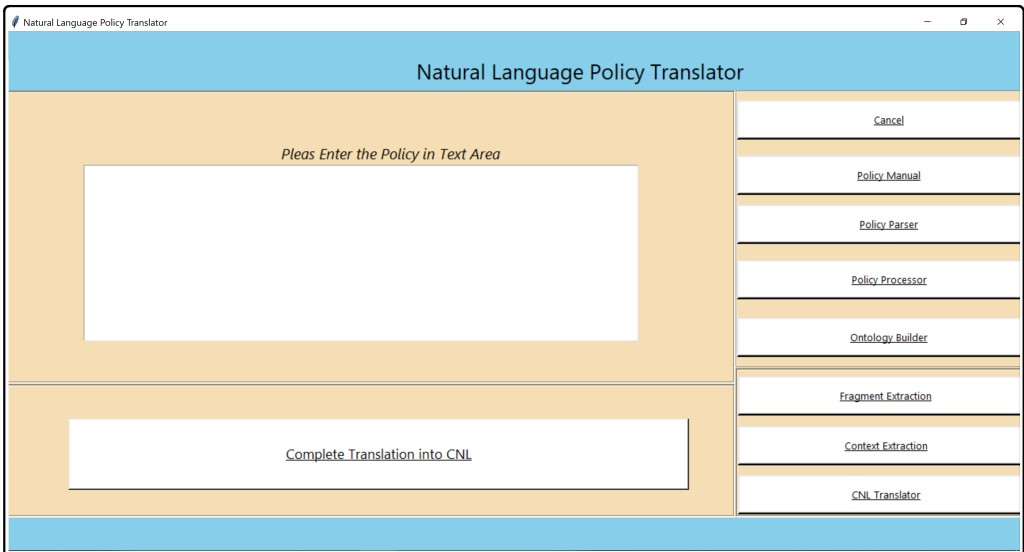

**Figure 8.** *NLPT 2.0* graphical user interface.

## 6. Experimental Social Networks Policies

The experimental setup was established based on the evaluation criteria to analyze the overall system performance. The components of the *Natural Language Policy Translator (CNLT) 2.0* are tested with five social networks' data privacy policies. The objective of the experiment is simply to check whether *Natural Language Policy Translator (NLPT) 2.0* is efficient enough to translate any common data privacy policy in the social network domain. To carry out the experiments, the subset of five different popular social network services' data privacy policies are gathered, i.e., Facebook [2], Twitter [3], Google [4], Instagram [12], and LinkedIn [13].

### 6.1. Experimental Setup

On social networking sites, there are different policy categories listed, e.g., Facebook (https://www.facebook.com/policies_center), Twitter (https://help.twitter.com/en), Google (https://policies.google.com/?hl=en-US), LinkedIn (https://www.linkedin.com/legal/user-agreement), Instagram (https://help.instagram.com/). The policies are manually skimmed, and only such policies that deal with user data management are considered. The choice of the policies is also determined by the fact that the policies are presented in full text, even if two different policies are defined in one paragraph. The subset of a data privacy policy from sets of policies is considered. In total, 100 different data privacy policies [53] that are related to user data regulation were chosen for the experiment.

The 'one output per input' evaluation paradigm is adopted to analyze components' accuracy using the following formula [54]:

$$A = \frac{\sum_{i=1..n,agr_i}}{n} = \frac{numbercount}{n} \tag{1}$$

where $agr\_i$ is 1 if $l_i = ti$ and 0 otherwise. Sometimes the inverse of accuracy, or error rate, is reported instead: $1 - A$. To determine the performances, the following evaluation criteria are established.

### 6.2. Evaluation Criteria

- How many policies are accurately parsed by *Policy Parser (PP)*, i.e., extracting action(s) and predicate(s) in a policy?
- How many times is a dictionary update required in terms of subject(s) and object(s) while processing the policy with *Policy Processor (PR)* until no update is required?

- How many times does an existing ontology have to be updated, or be created, while processing with *Ontology Builder (OB)* until no update is required?
- Can the *Fragment Extractor (FE)* accurately extract the fragment from the given policy?
- Can the *Context Extractor (CFE)* properly extract the context(s) from the input policy?
- Can the *Controlled Natural Language Translator (CNLT)* correctly classify the policy as authorization, obligation, and prohibition fragment?
- What is the success rate of *Policy Parser (PP)*, *Fragment Extractor (FE)*, *Context Extractor (CFE)* and *Controlled Natural Language Translator (CNLT)* over the total number of policies?

### 6.3. Experimental Operations

To maintain uniformity for components testing, the policies are processed in the following manner:

1. **Run Policy Parser**

   - Count the number of policies parsed correctly in a single iteration.

2. **Run Policy Processor**

   - Count the number of policies where it is required to update the vocabulary, with new subjects, objects, and predicates until no update is required.

3. **Run Ontology Builder**

   - Count the number of policies where it is required to create or update the ontology dictionary until no update is required.

4. **Run Fragment Extractor**

   - Count the number of fragments correctly extracted in a single iteration.

5. **Run Context Extractor**

   - Count the number of contexts correctly extracted in a single iteration.

6. **Run Controlled Natural Language (CNL) Translator**

   - Count the number of policies where a *Controlled Natural Language Translator (CNLT)* successfully identifies fragments as authorizations, obligations, and prohibitions in a single iteration.

Considering the experimental operations (Section 6.3), for example, the policy writer inputs a policy into the system and runs the policy parser in a single iteration. The obtained result from *Policy Parser (PP)* is analyzed to validate whether verbs and predicates are correctly identified, and the result is noted. Similarly, the same is performed for all other components. All *100* policies are processed according to the above-defined operations, and the obtained results for each component are stored, evaluated against the criteria defined in Section 6.2 and reported in Section 7.

The success rate of the following components, i.e., *Policy Parser (PP)*, *Fragment Extractor (FE)*, *Context Extractor (CFE)*, and *Controlled Natural Language Translator (CNLT)*, is calculated using the following formula:

**Success rate formula:**

$$\textbf{Success Rate} = \frac{X}{T} * 100$$

where:

$T$ = Total number of policies.

$X$ = Number of policies correctly parsed by *Policy Parser (PP)* or number of policiesaccurately extracted by *Fragment Extractor (FE)* and *Context Extractor (CFE)* in a single iteration or number of policies correctly recognized as authorization, obligation, and prohibition fragment by *Controlled Natural Language Translator (CNLT)*.

*6.4. Performance Evaluation Criteria*

Referring to the 'summarizing and comparing performance' evaluation paradigm [54], the performance metric scale for for Policy Parser (PP), Fragment Extractor (FE), Context Extractor (CFE) and Controlled Natural Language Translator (CNLT) is defined as reported in Table 1. The upper bound is set if the component's success rate is above or equal to 80% and the lower bound is above or equal to 20%.

**Table 1.** Performance evaluation against success rate.

| Success Rate | 80% >= | 80% < and >=60% | 60% < and >=40% | 40% < and >=20% | 20% < |
|---|---|---|---|---|---|
| Rating | Excellent | Above Average | Average | Below Average | Low |

## 7. Results and Discussion

Tables 2 and 3 show the results of each component's performance with respect to the evaluation criteria described in Section 6, where the first row denotes the evaluation criteria, the first column identifies the targeted social network domain, and the other columns offer information about the evaluation. The complete results of all *100* policy translations are available at [53].

**Success Rate:** For Google, *Policy Parser (PP)*, *Fragment Extractor (FE)*, *Context Extractor (CFE)* and *Controlled Natural Language Translator (CNLT)* success rates are calculated as follows:

$$PP \text{ Success Rate} = (14/20)*100 = 70\%$$

$$FE \text{ Success Rate} = (16/20)*100 = 80\%$$

$$CE \text{ Success Rate} = (13/20)*100 = 65\%$$

$$CNLT \text{ Success Rate} = (18/20)*100 = 90\%$$

The computation of the success rate for the rest of the social network policies is depicted in Figure 9.

The analysis of the obtained results is as follows:

- **Google:** Out of 20 policies, 14 *(70%)* are accurately parsed by the *Policy Parser (PP)*, identifying terms either as the main action or predicate(s). Initially, the dictionary has been updated 11 times *(55%)* with new terms for subjects and objects (including seven times *(35%)* with predicates due to the wrong parser tagging). The vocabulary for ontologies required an update eight times *(40%)* manually by the *Policy Writer*. It is hypothesized as if the percentage for dictionary and ontology updates is becoming low, as the system becomes more efficient at classifying subject(s), object(s), or predicates within a policy with less human intervention.
  In total, 16 fragments *(80%)* and 13 contexts *(65%)* have been properly extracted, while for 18 policies *(90%)*, the *Controlled Natural Language Translator (CNLT)* classified terms as either authorization, obligation, or prohibition fragments.
- **Facebook:** 13 policies *(65%)* are correctly parsed by *Policy Parser (PP)*, and the dictionary and ontology are updated by the *Policy Writer* 7 *(35%)* and 4 *(20%)* times. *Fragment Extractor (FE)* classifies 17 *(85%)* fragments and *Context Extractor (CFE)* 11 *(55%)* contexts correctly. *Controlled Natural Language Translator (CNLT)* validates 16 *(80%)* policy fragments as authorizations, obligations, or prohibitions accurately.
- **Twitter:** The obtained results of Twitter appear to be quite promising. In total, 16 *(80%)* policies are correctly parsed, while there was no need to update vocabularies for ontology and terms. *Fragment Extractor (FE)* obtains 20 *(100%)*) fragments accurately. A total of 16 *(80%)* contexts are correctly extracted by *Context Extractor (CFE)*. *Controlled Natural Language Translator (CNLT)* recognizes all 20 *(100%)* policies as authorization, obligation, or prohibition.

- **LinkedIn and Instagram:** For LinkedIn and Instagram, 17 *(85%)* and 16 *80* are properly parsed. The dictionary terms and ontologies are required to update only 1–2 times. *Fragment Extractor (FE)* and *Context Extractor (CFE)* obtained 18 *(90%)* fragments and 15 *(75%)* contexts for the premier, 20 *(100%)* and 15 *(75%)* for later. The performance of *Controlled Natural Language Translator (CNLT)* is *(95%)* for LinkedIn and *(100%)* for Instagram.

**Table 2.** Experiment results of social network data policies.

| Social Network Site | Total Policies | Correct Parsing | Accurate FE | Accurate CE | Distinction by CNLT |
|---|---|---|---|---|---|
| Google | 20 | 14 ($\approx$70%) | 16 ($\approx$80%) | 13 ($\approx$65%) | 18 ($\approx$90%) |
| Facebook | 20 | 13 ($\approx$65%) | 17 ($\approx$85%) | 11 ($\approx$55%) | 16 ($\approx$80%) |
| Twitter | 20 | 16 ($\approx$80%) | 20 ($\approx$100%) | 16 ($\approx$80%) | 20 ($\approx$100%) |
| LinkedIn | 20 | 17 ($\approx$85%) | 18 ($\approx$90%) | 15 ($\approx$75%) | 19 ($\approx$95%) |
| Instagram | 20 | 16 ($\approx$80%) | 20 ($\approx$100%) | 15 ($\approx$75%) | 20 ($\approx$100%) |
| Total | 100 | 76 ($\approx$76%) | 91 ($\approx$91%) | 71 ($\approx$70%) | 93 ($\approx$93%) |

**Table 3.** Experimental results of social network data policies.

| Social Network Site | Number of Policies | Dictionary Update | Ontology Update |
|---|---|---|---|
| Google | 20 | 11 ($\approx$55%) | 08 ($\approx$40%) |
| Facebook | 20 | 07 ($\approx$35%) | 04 ($\approx$20%) |
| Twitter | 20 | 0 ($\approx$0%) | 0 ($\approx$0%) |
| LinkedIn | 20 | 02 ($\approx$1%) | 01 ($\approx$1%) |
| Instagram | 20 | 01 ($\approx$1%) | 0 ($\approx$0%) |
| Total | 100 | 21 ($\approx$21%) | 13 ($\approx$12%) |

Initially, the *Natural Language Policy Translator (NLPT) 2.0* did not perform well parsing Google and Facebook policies, which also affected *Context Extractor (CFE)'s* performance. It is due to the *Policy Writer* finding difficulty in rephrasing policies. However, it is hypothesised that *Policy Writer* may gain the appropriate experience to input the policies correctly, which later happened in the case of Twitter, LinkedIn, and Instagram. This also reveals that the parser performance is relatively dependent on how the *Policy Writer* inputs policy.

The complete experimental results show that the whole performance of *Natural Language Policy Translator (NLPT) 2.0's* components is really good. In total, 76% of the policies are parsed precisely. Initially, the vocabulary for terms and ontology relations needs to be created or updated frequently by the *Policy Writer* due to new terms appearing in the policies. Once the dictionary is enriched enough with unique vocabularies and an ontologies relations lexicon, the human intervention decreases, and the system is trained enough to automatically tag the words as subject(s) and object(s) more accurately and produce proper ontology relations as in the case of Twitter, LinkedIn, and Instagram. The total accuracy of the *Fragment Extractor (FE)* and *Context Extractor (CFE)* is *(91%)* and *(71%)*, respectively. The overall accuracy achieved by the *Controlled Natural Language Translator (CNLT)* is *(93%)*.

The classification of *Policy Parser (PP)*, *Fragment Extractor (FE)*, *Context Extractor (CFE)*, and *Controlled Natural Language Translator (CNLT)* performance based on the established evaluation criterion in Table 1 is shown in Table 4. The performance of *Policy Parser (PP)* in the case of Google and Facebook is categorized as 'Above Average', while for other three, the performance appeared as 'Excellent'. The performance of *Fragment Extractor (FE)* and *Controlled Natural Language Translator (CNLT)* for all five social network policies is

classified as 'Excellent'. For *Context Extractor (CFE)*, in the case of Google, LinkedIn and Instagram, the performance appeared as 'Above Average', while for Facebook, 'Average', and 'Excellent' for Twitter. The complete performance of the *Fragment Extractor (FE)* and *Controlled Natural Language Translator (CNLT)* emerged as 'Excellent' for all five social network policies and 'Above average' for *Policy Parser (PP)* and *Context Extractor (CFE)*.

**Table 4.** Policy Parser (PP), Fragment Extractor (FE), Context Extractor (CFE) and Controlled Natural Language Translator (CNLT) performance evaluation against success rate.

| Case Study | PP Performance | FE Performance | CE Performance | CNLT Performance |
|---|---|---|---|---|
| Google | Above Average | Excellent | Above Average | Excellent |
| Facebook | Above Average | Excellent | Average | Excellent |
| Twitter | Excellent | Excellent | Excellent | Excellent |
| LinkedIn | Excellent | Excellent | Above Average | Excellent |
| Instagram | Excellent | Excellent | Above Average | Excellent |
| Total | Above Average | Excellent | Above Average | Excellent |

Fragment Extractor (FE); Context Extractor (CE); Controlled Natural Language Translator (CNLT).

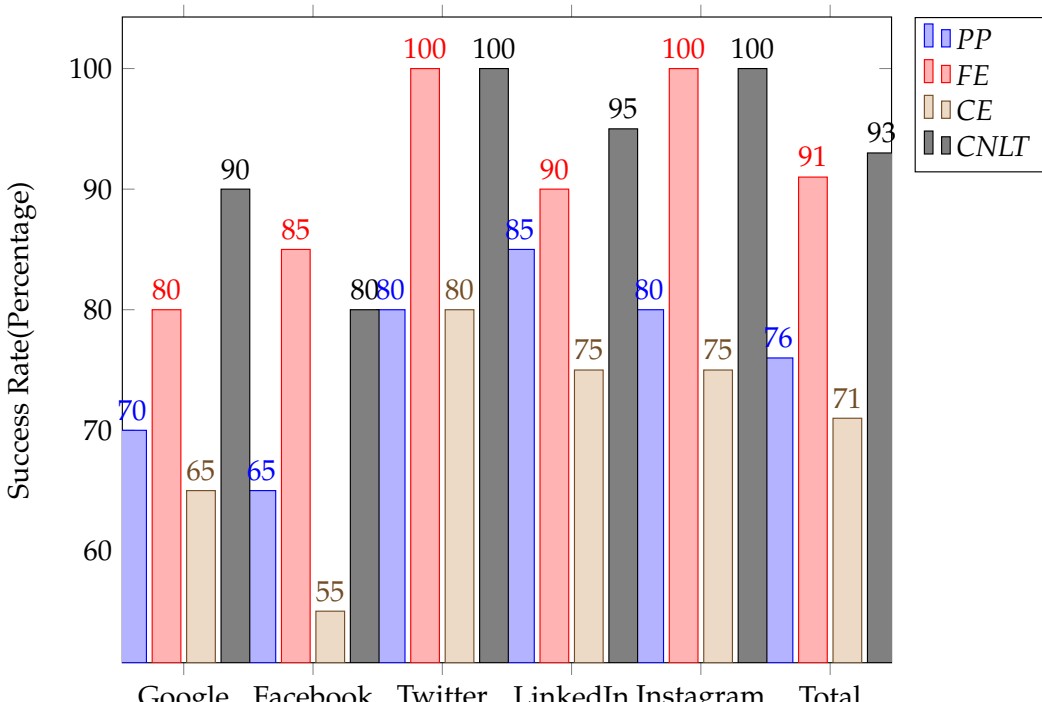

**Figure 9.** Performance of Policy Parser (PP), Fragment Extractor (FE), Context Extractor (CFE) and Controlled Natural Language Translator (CNLT) with respect to success rates.

During the experimentation, certain limitations and issues were observed. The policy writer found it quite difficult to rephrase the policy, which also impacts the performance of the *Policy Parser (PP)*. However, after gaining experience and the following instruction from the policy manual, it became easy to rephrase, and the *Policy Parsers (PP)* performance also improved, which can be observed in the case of Twitter, Instagram and LinkedIn, as shown in Table 2. The creation and updating of ontologies are currently manual, and we aim to make it automatic or semi-automatic in future work. The dependency parser tagging is based on how the policy is inputted. Wrong parsing may occur if the policies are composed of many complex sentences. The wrong parsing has a direct impact on the performance of

*Fragment Extractor (FE)* and *Context Extractor (CFE)*. The *Policy Writer* role has mitigated this issue at a certain level.

Moreover, automatic co-reference detection is not considered, and it will be assailed in future work. In addition, the complete process of translation is semi-automatic since it requires human intervention at different stages. However, it is argued that once the system is trained enough on upgraded lexicons with unique terms and ontology relations, the human interventione intervention lessens. In general, the overall performance indicators of the system's components indicate that *Natural Language Policy Translator (NLPT) 2.0* is capable of automatically translating any social network data privacy policy into a CNL4DSA.

## 8. Conclusions

The data management regulations on social media platforms are defined using Natural Language (NL). These policies are not machine-checkable and are mostly vague and imprecise. The tight lexicon(s), vocabulary, syntax(es), grammatical structure, and pragmatics of Controlled Natural Languages CNLs make them more suitable for machine processing. This study proposes a unique technique for handling data privacy policies by introducing the translation of their natural language descriptions into phrases that are easier for non-experts in terms of readability and usability. The goal of this work is to provide an effective and efficient approach for the formal analysis of privacy rules. The aimed language for translation is Controlled Natural Language for Data Sharing Agreement [55], which is more similar to a pure Natural Language (NL) and is easier for non-experts in terms of understandability and usability [51].

The proposed methodology is based on processing the data policies through Natural Language Processing (NLP) techniques, logic programming and ontologies, recognising—and appropriately relating—the typical elements of a privacy policy in the original natural language statements. The prototype system that translates the data privacy policies is presented and is referred to as '*Natural Language Policy Translator (NLPT) 2.0*', an extended version of the system 'Natural Language Policy Translator (NLPT) 1.0' [6]. *Natural Language Policy Translator (NLPT) 2.0* is composed of different components that allow the end-user to analyze the functionality and operations required for policy translation. The system is equipped with a user-friendly GUI that supports end-users entering the policies, and the system translates it into CNL4DSA.

In conclusion, *Natural Language Policy Translator (NLPT) 2.0* provides the following functionalities:

- Parses the policy to extract action(s) and predicate(s).
- Processes policy by means of tokenization of sentences, words, stop-words, unigram tagging to label subject(s) and object(s) and extract subject(s), object(s), action(s) and predicate(s).
- Identifies and produces ontologies with respect to subject hasRole, the object hasCategory, the object hasPurpose, relation as subject predicate object, etc.
- Extracts the fragment from the original policy.
- Extracts the context from the original policy.
- Classifies between authorization, obligation, and prohibition fragments.
- Performs all the above tasks separately and together.

For experimentation, *NLPT 2.0* is tested with five popular social network data privacy policies. To analyze the components' performance, a criterion is defined for evaluation. The obtained results of various components' performances are between 70% and 95%. The performance indicators of the system components appeared relatively encouraging and satisfactory. Initially, the work aims to provide complete automatic machine translation without human intervention. However, due to certain limitations found during experimentation, there is a need for human intervention to define the vocabulary for unigram tagging and ontologies at the initial stage. It is argued that once the system is fully trained with unique terms for the vocabulary and ontology lexicon, human intervention becomes

minimal, and the system can automatically tag the words as subject(s) and object(s) more accurately and produce proper ontology relations.

As discussed in Section 7, the policy cannot simply be input in its original form since the dependency parser [41] may not be able to correctly parse too complicated phrases. Of course, improper parsing also has a direct influence on Fragment Extractor (FE) and Context Extractor (CFE) performance. This issue has been dealt with by introducing the role of a *Policy Writer* to enter the policy following the manual's guidelines. The approach is currently not fully automatic because it needs human involvement at certain points, yet it is close to fully automatic translation. All these issues will be addressed in future work.

**Author Contributions:** Conceptualization, I.K.T.; methodology, I.K.T.; software, I.K.T. and I.A.; validation, F.J. and N.Y.; writing—original draft preparation, I.K.T.; writing—review and editing, I.A. and F.J.; visualization, N.Y.; supervision, I.A.; funding acquisition, N.Y. All authors have read and agreed to the published version of the manuscript.

**Funding:** This research received no external funding.

**Institutional Review Board Statement:** Not applicable.

**Informed Consent Statement:** Not applicable.

**Data Availability Statement:** The complete translation of all 100 policies is available here [53].

**Acknowledgments:** The authors would like to extend their gratitude to Marinella Petrochhi, Asif Khalid, Khalid Rasheed and Yaseen Khan Tanoli who helped in this study.

**Conflicts of Interest:** The authors declare no conflict of interest.

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
