# Peer review of "Systematic Machine Translation of Social Network Data Privacy Policies"

_applsci, doi:10.3390/app122010499_

Round 1

Reviewer 1 Report

In this paper, a unique technique to handling data privacy policies is described by presenting the translation of data privacy policies’ NL descriptions into CNL phrases, or CNL4DSA. It provides an effective and efficient method for the formal analysis of privacy rules. The paper is written well. However, there are still some problems. The following comments could be taken into account to improve the paper further.

1. Authors should pay attention to the use of tenses.

1) The summary of the research content in the Conclusion part needs to use the perfect tense.

2) The present tense should be used when introducing existing works.

2. Authors should pay attention to the detailed specifications of the paper.

1) Some figures are too blurry, such as Figures 2 and 5.

2) There are some garbled characters in this paper, such as lines 497, 498, and 647.

3. The experiments in this manuscript are not sufficient, more comparative experiments with existing techniques are expected.

4. The presentation could be significantly improved, e.g., the wrong use of articles and misspellings.

5. Please unify the format of the references part. The authors can refer to the format of the references in the published papers of the journal to be submitted.

Author Response

We thank the Associate Editor and the reviewers for their insightful and helpful comments on our manuscript. We have carefully considered their feedback and revised the paper, considering their suggestions and remarks. For their convenience, we have retyped the remarks in bold italics followed by our responses in a regular font in this document. Overall, we are incredibly grateful for the extensive review comments, which have provided invaluable guidance for the thorough revision and improvement of this manuscript.

Reviewer 1: Summary Remarks

In this paper, a unique technique to handling data privacy policies is described by presenting the translation of data privacy policies’ NL descriptions into CNL phrases, or CNL4DSA. It provides an effective and efficient method for the formal analysis of privacy rules. The paper is written well. However, there are still some problems. The following comments could be taken into account to improve the paper further.

  1. Authors should pay attention to the use of tenses.

Response: The paper has been proof readed again by authors and Grammarly paid version is used for this purpose.

  • The summary of the research content in the Conclusion part needs to use the perfect tense.

Response: The paper has been proof readed again by authors and one of the author revised the section

  • The present tense should be used when introducing existing works.

Response: The paper hase been proof readed again by authors and one of the author revised the section and track changes has been applied in the manuscript.

  1. Authors should pay attention to the detailed specifications of the paper.

Response: The paper hase been proof readed again by authors.

1) Some figures are too blurry, such as Figures 2 and 5.

Response: Both Figures are enhased and are more readable now.

  • There are some garbled characters in this paper, such as lines 497, 498, and 647.

Response: The paper hase been proof readed again by authors and garbled chracterers including above mentioned in the lines are removed.

  1. The experiments in this manuscript are not sufficient, more comparative experiments with existing techniques are expected.

Response: The experiments sectin (section 06) explain the choice of policies, how and which policies are chosen. The section result show the comparative output in details. The authors try their best to explained in a well manner.

  1. The presentation could be significantly improved, e.g., the wrong use of articles and misspellings.

Response: The paper hase been proof readed again by authors to remove the wrong use of articles and misspellings

  1. Please unify the format of the references part. The authors can refer to the format of the references in the published papers of the journal to be submitted.

Response: All refererence are unified according to journal standard.

Reviewer 2 Report

This paper proposes an approach to systematically translate privacy statements related to data from NL into a controlled natural one, CNL4DSA, to improve machine processing. The methodology combines standard Natural Language Processing techniques, logic programming, and ontologies. The proposed technique is demonstrated with a prototype implementation and tested with policy examples. The system experimented with some data policies from five different social network service providers.

The paper addresses an interesting question that has yet to receive great responses from the community in recent years. The interpretation of policies in an automatic way is something that is still in the investigation phase. Therefore, all hypotheses of possible solutions to the problem must be studied and analyzed to obtain a possible ideal solution. Understanding the structure of machine-interpreted language can take time to approach, master and develop. In this way, if the process is streamlined, it becomes an asset for the understanding, use and development of technology, having, as a practical effect, better protection of the user's privacy.

Regarding the writing style, the author makes extensive use of abbreviations, which in turn, due to the quantity, makes the reading experience more difficult as the reader has to go back to previous sections of the text to remember the meaning of the abbreviation. The author also uses a combination of words using an apostrophe (e.g., don’t, can’t), which is inadequate for a technical and scientific document. It also has some punctuation errors, namely the use of ":" at the end of paragraphs when a new section follows, among other situations (e.g., before 5.2, lines in sequences 632,633 and 635, or even in line 641). Regarding the structure, the document presents some reference failures in the final sections.

Regarding state of the art, it seems well built and with a wide range of references and works developed on the subject. A vast temporal criterion for references is evident (the oldest being dated from 1994). This requires a brief justification. It is also noted that there has yet to be a final evaluation or comment on the current state of development of the topic/scientific area. In Section 3, the expression "methodological approach" should be modified to something like "design approach". The term methodological, regarding research, is usually connected to a well-known set of methods used to conduct research, not the design of a solution.

Regarding the proposed solution, the prototype presented and the validation denoted an effort of practical application since the authors use privacy policies of 5 primary social network services used daily by millions of people worldwide. However, at the beginning of the paper, there is an expectation that the system used will do all the analysis work automatically. This expectation is quickly abandoned when we look at Figure 1 and realize that human intervention is necessary throughout the various phases of the development process. Although the authors assume in the conclusion that the system is semi-automatic, reviewing the abstract and introduction would be positive in order not to give the idea that the system is entirely automatic. Concerning validation, it is necessary to clarify the criteria used in the choice of samples, not only in quantity (why 20 from each social network?) but also in terms of the type of policy used and why. It would be best if you focused more on those policies and the possible restrictions on their interpretation by the machine.

Author Response

Response to Review Comments

" Systematic Machine Translation of Social Network Data Privacy Policies"

We thank the Associate Editor and the reviewers for their insightful and helpful comments on our manuscript. We have carefully considered their feedback and revised the paper, considering their suggestions and remarks. For their convenience, we have retyped the remarks in bold italics followed by our responses in regular font in this document. Overall, we are incredibly grateful for the extensive review comments, which have provided invaluable guidance for the thorough revision and improvement of this manuscript.

Reviewer 2: Summary Remarks

This paper proposes an approach to systematically translate privacy statements related to data from NL into a controlled natural one, CNL4DSA, to improve machine processing. The methodology combines standard Natural Language Processing techniques, logic programming, and ontologies. The proposed technique is demonstrated with a prototype implementation and tested with policy examples. The system experimented with some data policies from five different social network service providers.

The paper addresses an interesting question that has yet to receive great responses from the community in recent years. The interpretation of policies in an automatic way is something that is still in the investigation phase. Therefore, all hypotheses of possible solutions to the problem must be studied and analyzed to obtain a possible ideal solution. Understanding the structure of machine-interpreted language can take time to approach, master, and develop. In this way, if the process is streamlined, it becomes an asset for the understanding, use, and development of technology, having, as a practical effect, better protection of the user's privacy.

Response: As discussed in Section 7: Result and discussion and Section Conclusion:

“Initially, the work aims to provide complete automatic machine translation without human intervention. However, due to certain limitations found during experimentation, there is a need for human intervention as defining vocabulary for unigram tagging and ontologies at the initial stage. It is argued that this human intervention becomes minimal once the system is fully trained with unique terms vocabulary and ontologies lexicon, and the system is trained enough to automatically tag the words as subject(s), and object(s) more accurately and produced proper ontology relations as in case of Twitter, LinkedIn, and Instagram.
“. Moreover the translation into CNL4DSA is quite helpful for users to understand their privacy issues and as well as for machine translation.

Regarding the writing style, the author makes extensive use of abbreviations, which in turn, due to the quantity, makes the reading experience more difficult as the reader has to go back to previous sections of the text to remember the meaning of the abbreviation. The author also uses a combination of words using an apostrophe (e.g., don’t, can’t), which is inadequate for a technical and scientific document. It also has some punctuation errors, namely the use of ":" at the end of paragraphs when a new section follows, among other situations (e.g., before 5.2, lines in sequences 632,633, and 635, or even in line 641). Regarding the structure, the document presents some reference failures in the final sections.

Response: The paper has been proofread again by the authors and one of the authors revised the section and track changes have been applied in the manuscript. Also, all references are unified according to journal standards.

Regarding the state of the art, it seems well built and with a wide range of references and works developed on the subject. A vast temporal criterion for references is evident (the oldest being dated from 1994). This requires a brief justification. It is also noted that there has yet to be a final evaluation or comment on the current state of development of the topic/scientific area. In Section 3, the expression "methodological approach" should be modified to something like "design approach". The term methodological, regarding research, is usually connected to a well-known set of methods used to conduct research, not the design of a solution.

Response: As per the reviewer's suggestion, the methodological approach has been replaced with the design approach. In terms of state of art, it is explained in the manuscript that The approach proposed by Tanoli et al.~\cite{tanoli2018towards}, is one of the few initiatives to bridge the gap between modifying and processing a \chreplaced{Controlled Natural Language (CNL)}{CNL}, like CNL4DSA, and maintaining complete readability of privacy policies by expressing them in a natural language.

Regarding the proposed solution, the prototype presented and the validation denoted an effort of practical application since the authors use privacy policies of 5 primary social network services used daily by millions of people worldwide. However, at the beginning of the paper, there is an expectation that the system used will do all the analysis work automatically. This expectation is quickly abandoned when we look at Figure 1 and realize that human intervention is necessary throughout the various phases of the development process. Although the authors assume in the conclusion that the system is semi-automatic, reviewing the abstract and introduction would be positive in order not to give the idea that the system is entirely automatic. Concerning validation, it is necessary to clarify the criteria used in the choice of samples, not only in quantity (why 20 from each social network?) but also in terms of the type of policy used and why. It would be best if you focused more on those policies and the possible restrictions on their interpretation by the machine.

Response: As explained in the previous response and already discussed in Section 7: Result and discussion and Section Conclusion:

“Initially, the work aims to provide complete automatic machine translation without human intervention. However, due to certain limitations found during experimentation, there is a need for human intervention as defining vocabulary for unigram tagging and ontologies at the initial stage. It is argued that this human intervention becomes minimal once the system is fully trained with unique terms vocabulary and ontologies lexicon, and the system is trained enough to automatically tag the words as subject(s), and object(s) more accurately and produced proper ontology relations as in case of Twitter, LinkedIn, and Instagram.
“. Moreover the translation into CNL4DSA is quite helpful for users to understand their privacy issues and as well as for machine translation.

Regarding the choice of 20 policies, The experiments section (section 06) explains the choice of data policies, and how and which policies are chosen. We only targeted such policies that deal with user data management. The section result shows the comparative output in detail. The authors try their best to explain the results in a good manner.
